# Outpatient Antibiotic and Antiviral Utilization Patterns in Patients Tested for Respiratory Pathogens in the United States: A Real-World Database Study

**DOI:** 10.3390/antibiotics11081058

**Published:** 2022-08-04

**Authors:** Jenny Tse, Aimee M. Near, Mindy Cheng, James Karichu, Brian Lee, Susan N. Chang

**Affiliations:** 1IQVIA, Cambridge, MA 02139, USA; 2Roche Diagnostics Solutions, 4300 Hacienda Drive, Pleasanton, CA 94588, USA

**Keywords:** antibiotic, antiviral, stewardship, diagnostics, respiratory tract infections, outpatients, real-world

## Abstract

This retrospective observational study evaluated outpatient treatment patterns among patients with molecular-based viral diagnostic testing for suspected upper respiratory tract infections in the United States. Patients with a respiratory viral test were identified from 1 August 2016 to 1 July 2019 in a large national reference laboratory database linked to IQVIA’s prescription and medical claims databases. Antibiotic and influenza antiviral treatment patterns were reported up to 7 days post-test result. Predictors of antibiotic utilization were assessed using multivariable logistic regression. Among 9561 patients included in the study, 24.6% had evidence of ≥1 filled antibiotic prescription. Antibiotic utilization was higher in patients who tested negative for all viral targets (odds ratio [OR], 1.33; 95% confidence interval [CI], 1.17–1.50) and patients positive for non-influenza viruses (OR, 1.28; 95% CI, 1.09–1.51) compared with those influenza-positive only. Age ≥ 50 years and location outside of the northeast United States also predicted antibiotic utilization. Influenza antivirals were more common in influenza-positive patients compared with patients with other test results (32.5% vs. 3.6–9.0%). Thus, in this real-world study, antibiotic utilization was elevated in patients positive for non-influenza viruses, although antibiotics would generally not be indicated. Further research on pairing diagnostic tools with outpatient antibiotic stewardship programs is needed.

## 1. Introduction

Upper respiratory tract infections are common, with an estimated 500 million non-influenza respiratory infections in the United States (US) every year [1]. Despite acute respiratory infections being predominantly viral in origin, they are the most common diagnosis for which antibiotics are prescribed [2]; up to 40% of these prescriptions are considered unnecessary [3,4,5]. The overuse of antibiotics is a key contributor to antibiotic resistance; therefore, antibiotic stewardship programs are a public health priority [6,7]. The U.S. National Action Plan for Combating Antibiotic-Resistant Bacteria (CARB) for 2020–2025 includes an objective to improve national outpatient antibiotic utilization, by gaining a greater understanding of trends in unnecessary antibiotic prescribing patterns [8].

One aspect of antibiotic stewardship is diagnostic stewardship, ordering the right test to inform timely and appropriate test-guided treatments and improve patient care [9,10]. In the last decade, dozens of highly sensitive molecular-based diagnostic tests for respiratory pathogens have been cleared or approved by the U.S. Food and Drug Administration [11,12]. These tests can provide results for one or more pathogens relatively rapidly, within minutes or hours [12]. Several studies in adult and pediatric populations have reported that rapid, accurate test results may impact treatment decision-making in hospital settings, including improving influenza antiviral prescribing in patients who tested positive for influenza [13,14,15] and reducing the duration of antibiotics in patients with positive influenza or other viral tests [16,17]. However, other studies have found that test results may not reduce the frequency of antibiotic prescribing [13,18].

Despite our current repository of tools, including antibiotic stewardship programs and diagnostic testing, judicious antibiotic utilization remains suboptimal. In addition, the impact of molecular testing on treatment decisions is unclear, especially in the outpatient setting. This retrospective database study aims to describe outpatient antibiotic and influenza antiviral treatment patterns following molecular-based tests for acute respiratory tract infections and to explore predictors of antibiotic utilization in clinical practice.

## 2. Results

In total, 9561 patients received a respiratory viral test of interest in the large national reference laboratory database linked to IQVIA′s prescription claims (LRx) and medical claims (Dx) databases and met all study selection criteria (Figure 1). The demographic and baseline clinical characteristics of the cohort are described in Table 1. The mean age was 36.1 years (standard deviation [SD], 30.8 years), with 45.3% of the cohort aged 0–17 years and 30.7% aged 65 years or older. The cohort was predominantly female (58.8%), and most patients were located in the Northeast or Southern United States (43.1% and 33.2%, respectively). Nearly all patients were either commercially insured (70.6%) or had Medicare (29.3%). The majority of the cohort had a baseline Charlson Comorbidity Index (CCI) score of 0 (69.1%), and the most frequently observed comorbidities were chronic pulmonary disease (12.4%), asthma (8.9%), and mild to moderate diabetes (7.4%).

Most of the index viral tests were ordered during an influenza season (91.9%), with 46.2% of all tests ordered by primary care physicians and 34.6% ordered by pediatricians (Table 2). The majority (59.1%) of patients had a respiratory-related diagnosis code on the index date; the most common diagnoses were unspecified upper respiratory tract infection (22.1%), influenza (18.7%), and pharyngitis (16.6%). Of the respiratory-specific diagnoses, 36.1% had tier 3 diagnoses where antibiotics were not indicated, 20.8% had tier 2 diagnoses where antibiotics would potentially be indicated, and 2.7% had tier 1 diagnoses where antibiotics would almost always be indicated.

In total, 57.5% of the cohort tested negative for all viral targets, 22.5% tested positive for influenza, 20.8% tested positive for a non-influenza viral target, and 0.7% of patients were co-infected with influenza and a non-influenza virus. For each specific viral target, the highest positivity rates were observed for the combination rhinovirus/enterovirus test (25.2%), rhinovirus (22.7%), unspecified coronavirus (20.0%), and influenza A (14.9%) (Appendix A).

Utilization of antibiotics and influenza antivirals in the cohort stratified by test result is described in Figure 2. Overall, 24.6% of the cohort had ≥1 antibiotic pharmacy claim, including patients with antibiotics only and those with both antibiotics and influenza antivirals. The highest antibiotic utilization was observed among the subgroup who tested negative for all viral targets (26.6%), followed by patients who tested positive for any non-influenza target (22.8%), and followed by patients who tested positive for influenza (20.4%). Among patients who tested positive for non-influenza viruses, the highest rates of antibiotic utilization were among patients who tested positive for respiratory syncytial virus (RSV; 28.4%) and human metapneumovirus (23.2%). Influenza antivirals were observed in 13.1% of the overall cohort (including patients with influenza antiviral only and those with both antibiotics and influenza antivirals) and the highest utilization was observed among patients who tested positive for influenza (32.5%), followed by patients who tested negative for all targets (9.0%). Additionally, of the patients who tested positive for influenza, 22.83% had ≥1 claim for influenza antiviral only and 8.64% had ≥1 claim for both an antibiotic and influenza antiviral.

In the logistic regression model among all patients, several characteristics were identified as significantly increasing the odds of antibiotic utilization (Table 3). Specifically, the odds of antibiotic utilization were 27–40% higher among patients aged 50 years or older compared with those aged 18–49 years old, 34–63% higher among patients located in U.S. states outside of the Northeast, 50–70% higher among patients who had tier 1 or tier 2 respiratory-specific diagnosis codes on the index date compared with patients with tier 3 diagnoses, and 28–33% higher among patients who tested positive for a non-influenza virus or who were co-infected with influenza and non-influenza virus compared with patients who tested positive for influenza only. Only a younger age (5–17 years) was associated with lower odds of antibiotic utilization.

## 3. Discussion

In this real-world study of patients with suspected upper respiratory tract infections, outpatient antibiotic prescribing was common and observed in about one-quarter of all patients who received molecular respiratory viral testing. Overall, we did not find a strong association between positive viral test results, particularly for non-influenza viruses, and a reduction in antibiotic prescribing. Compared with patients who only tested positive for influenza, those who tested positive for non-influenza viruses or negative for all viral targets had 28–33% higher odds of antibiotic utilization. Antibiotics are generally not indicated for the treatment of respiratory viruses or in the absence of an identified pathogen; therefore, this was a surprising finding. However, these results may be explained by several unmeasured factors. First, the antibiotic treatment decisions may have been informed by bacterial test results that were not observable in this study or the clinical suspicion of bacterial infection. This scenario seems probable based on our analysis of respiratory-related diagnosis codes, where patterns in the odds of antibiotic utilization matched up with the diagnosis tiers in the way one might expect (i.e., highest odds for patients with tier 1 diagnoses where antibiotics would almost always be indicated, and then tier 2 diagnoses, compared with tier 3), which suggests that the clinicians’ diagnoses may have warranted an antibiotic in some cases. Second, the timing and speed of the test results may have also played a role in impacting decision-making. In a retrospective study conducted by Rogers et al. of children hospitalized with an acute respiratory tract infection, the duration of antibiotic treatment was reduced by half a day if the positive test result was reported within 4 h compared with longer than 6 h [17]. The importance of timing may also explain why only about one-third of patients in this study had evidence of influenza antivirals following a positive influenza test result; because antivirals should ideally be initiated within 48 h of symptom onset [19], we may expect that some patients filled a prescription for an influenza antiviral before the tests results were available and others were outside this treatment window by the time of the test result. Third, we can consider the study findings in the context of the outpatient antibiotic stewardship ecosystem and the distinction between information and implementation; although a test result is available, it will not necessarily be used by the provider. Several environmental and resource barriers to the implementation of antibiotic stewardship programs have been identified in surveys of providers, infectious disease specialists, hospital representatives, and special interest groups, including a lack of key staffing support, inadequate data and information systems, and inadequate financial resources [20], all of which can impede the usability of a positive diagnostic test result. As such, more attention to addressing these barriers towards optimization of outpatient antimicrobial stewardship is needed. Moreover, a 2020 study indicated that physicians frequently experience pressure to prescribe antibiotics from patients or their parents [21], which suggests that inappropriate antibiotic prescribing could be occurring to appease patients.

Nevertheless, our findings of treatment patterns in the outpatient setting are consistent with previously conducted retrospective studies in inpatient settings, emergency departments, and hospital outpatient clinics, which suggest that respiratory viral testing alone may not be sufficient for improving appropriate antibiotic utilization [15,16,18]; rather, the combination of diagnostic testing and antimicrobial stewardship tools is needed. Several studies have suggested diagnostic tools, including rapid blood culture molecular tests [22,23], procalcitonin tests [24], and chest imaging [25] in hospital or emergency department settings may provide an added benefit when combined with antimicrobial stewardship programs. Guidance on how to optimize the utilization and interpretation of diagnostic tests through antimicrobial stewardship programs is needed in outpatient settings, where research has been sparse.

In addition, this study investigated the association between patient characteristics and antibiotic utilization, which may help inform targeted outpatient antibiotic stewardship efforts. We observed that older adults ≥50 years of age had 27–40% higher odds of antibiotic treatment compared with younger adults, which aligns with prior studies of patients treated for respiratory tract infections in the emergency department setting [26,27]. In addition, patients located outside of the northeastern United States had 34–63% higher odds of antibiotic utilization compared with patients in the Northeast. Geographic variation in outpatient antibiotic prescribing has previously been reported [28,29], but the regions with the highest antibiotic utilization varied across studies and may be a reflection of the differences in the underlying study populations (i.e., Medicare beneficiaries [29] and commercially insured individuals [28]) and the granularity of the regions reflected (i.e., U.S. census regions [28,29] and U.S. states [30]).

Although there was no strong association between positive viral test results and antibiotic prescribing, we observed a favorable trend in higher antiviral utilization in the presence of a positive influenza test result (32.5%) compared with patients who tested positive for any non-influenza virus (3.6%) or negative for all viruses (9.0%). These findings of increased influenza antiviral use following testing have also been demonstrated in prior studies [13,14,15], which suggests that positive viral test results can enable the prompt and appropriate initiation of influenza antivirals. Nonetheless, the proportion of influenza-positive patients treated with influenza antivirals was lower than anticipated and may be related to the delay from symptom onset to testing and to the receipt of the test result, given that antiviral therapy is associated with better outcomes when initiated within 2 days of symptom onset [31]. The importance of timely, accurate results in informing patient management with influenza antivirals draws parallels with new COVID-19 antiviral drugs that must be given early in the course of infection (i.e., within 3 to 5 days of symptom onset) [32,33] and concerns about sufficient access to timely testing and optimized treatment decisions in real-world settings [34,35]. Additionally, there have been studies suggesting high empiric antibiotic utilization in COVID-19 patients, despite a lack of evidence of bacterial co-infection [36,37]. As such, the importance of highly sensitive and specific molecular tests to improve patient management and antibiotic utilization warrants further investigation both outside and within the current context of the COVID-19 pandemic.

The key strength of this study is the use of a nationally representative laboratory database linked to large pharmacy and medical claims databases in the United States, which allows for a comprehensive view of outpatient antibiotic and influenza antiviral utilization following receipt of a molecular-based viral test. Other studies examining the association between the results of viral testing for respiratory tract infections and treatment outcomes have typically used data from inpatient, emergency department, and urgent care settings in single institutions or limited analyses to pediatric or adult populations, whereas our study cohort reflects a variety of age categories, geographies, and insured populations. Given the majority of antibiotic expenditures occur in the outpatient setting [38] and the need to improve outpatient antibiotic utilization in the United States [8], this study also fills a gap in the current research by characterizing treatment patterns in patients from a variety of outpatient settings. However, several limitations relevant to claims database studies must be noted. First, because the treatment patterns reported in this study were based on prescription fills, we cannot confirm whether the patients took the antibiotic or influenza antiviral medication after picking it up from the pharmacy, or if they were later instructed to discontinue treatment by their providers. Second, although the prescription fills occurred on or after the viral test result became available, it is possible that the prescriptions may have been written at an earlier time, and treatment initiation was not informed by the test results. Further research examining the clinical utility of a diagnostic test result, beyond guiding treatment initiation, is warranted. For example, changes in treatment plans (e.g., the discontinuation or de-escalation of antibiotics) and antibiotic treatment duration may be informed by diagnostic testing. Third, clinical data from bacterial test results were not available to confirm whether antibiotic utilization was justified. Although we limited the timeframe for reporting antibiotic utilization to the 7-day period after the test result, it is possible that the antibiotic was prescribed to treat a condition unrelated to the suspected respiratory tract infection. To address these limitations, we evaluated tiers of respiratory-related ICD-10 diagnosis codes present on the index date and found that higher tiers were indeed associated with increased odds of antibiotic utilization, suggesting that antibiotic utilization may have been justified. We incorporated these tiers into our adjusted analyses to better account for unmeasured factors related to clinical decision-making. Despite these limitations, this study provides novel insight into patterns of diagnostic testing and antibiotic utilization in the outpatient setting which can inform antimicrobial stewardship efforts in the future.

## 4. Methods

### 4.1. Study Population

This retrospective observational study used data from a large national reference laboratory linked to IQVIA’s prescription claims (LRx) and medical claims (Dx) databases during the study period from 1 February 2016 to 31 July 2019 (latest data available at time of analysis). The laboratory database consists of over one decade of laboratory data for approximately 150 million patients and is nationally representative of the U.S. population. Key information in the laboratory database includes patient demographics (age, sex, and geography), diagnostic tests identified using Logical Observation Identifiers Names and Codes (LOINC) and lab test names, and test results. The LRx database captures information on dispensed prescriptions with 92% coverage of prescriptions from the retail channel, 72% coverage of standard mail service, and 76% coverage of long-term care facilities. The Dx database captures over one billion pre-adjudicated claims and three billion records obtained annually from approximately 800,000 office-based physicians and specialists, with 75% of American Medical Association providers captured. Medical claims from ambulatory and general health care sites (as well as outpatient clinics associated with hospitals such as rehabilitation, same day surgery, and chemotherapy centers) are also included in the Dx database. All data are Health Insurance Portability and Accountability Act (HIPAA)-compliant to protect patient privacy. The study dataset was created based on an HIPAA-compliant linking process using IQVIA’s patented and proprietary encryption algorithm [39,40].

Inclusion and exclusion criteria for patient selection are described in Figure 1. Briefly, patients with ≥1 claim with a Current Procedural Terminology (CPT) code for a respiratory viral test of interest (CPT 87502, 87631, 87632, or 87633) in Dx during the selection window from 1 August 2016 to 1 July 2019, with a corresponding record for the test in the laboratory database based on test order codes or order names were included in the study. The tests of interest included any of four polymerase chain reaction (PCR)-based tests that included influenza as a target, at a minimum. The date of the earliest claim for a test of interest in Dx was considered as the index date. All patients were required to have patient eligibility in LRx and Dx and pharmacy stability in LRx during the 6-month pre-index (i.e., baseline) and minimum 1-month post-index (i.e., follow-up) periods.

### 4.2. Study Measures

For the cohort of patients meeting all selection criteria, patient demographics and baseline clinical characteristics (including CCI and select comorbidities) were evaluated during the 6-month baseline period, and characteristics of the index test (including timing of the index test, ordering physician specialty, presence of respiratory-related diagnosis codes on index date, and test results) were reported from data on the index date. Respiratory-related International Classification of Diseases Tenth Revision Clinical Modification (ICD-10-CM) diagnosis codes were grouped into three tiers based on previous publications [2,5,41], where tier 1 included diagnoses for which antibiotics are almost always indicated (i.e., pneumonia), tier 2 included diagnoses for which antibiotics may be indicated (i.e., suppurative otitis media, pharyngitis, sinusitis, tonsillitis), and tier 3 included diagnoses for which antibiotics are not indicated (i.e., acute bronchitis, acute nasopharyngitis, influenza, laryngitis/tracheitis, non-suppurative otitis media, and unspecified upper respiratory tract infection); higher tiers were prioritized if patients had diagnosis codes in more than one tier. Positivity rates for each virus were also reported and were calculated as the number of patients with a positive test result divided by the number of patients tested for the specific virus.

Follow-up measures included antibiotic utilization and influenza antiviral utilization, which were reported in the overall cohort stratified by three non-mutually exclusive categories of test results (i.e., positive for influenza, positive for non-influenza virus, or negative for all test targets). Utilization was determined based on the presence of ≥1 pharmacy claim for an antibiotic or influenza antiviral picked up from a pharmacy on or up to 7 days after the index test result date.

### 4.3. Statistical Analysis

Logistic regression was used to estimate the odds ratios (ORs) and 95% confidence intervals (CIs) for the association between patient demographic and baseline clinical characteristics and characteristics of the index test (including test result and respiratory-related diagnosis code tier on the index date). *p*-values < 0.05 were considered statistically significant. All analyses were conducted using SAS version 9.4 (SAS Institute, Inc., Cary, NC, USA).

## 5. Conclusions

In summary, treatment with antibiotics for suspected respiratory tract infection in the outpatient setting was common in this large, real-world retrospective study. Although viral molecular testing had a demonstrated impact on patient management with influenza antivirals, its value on outpatient antibiotic utilization remains a topic for further exploration. Respiratory viral testing alone may not be sufficient for improving antibiotic utilization; expanding outpatient antimicrobial stewardship tools and further investigation into the barriers and facilitators of linking diagnostic test results with treatment decision-making are needed.

## Figures and Tables

**Figure 1 antibiotics-11-01058-f001:**
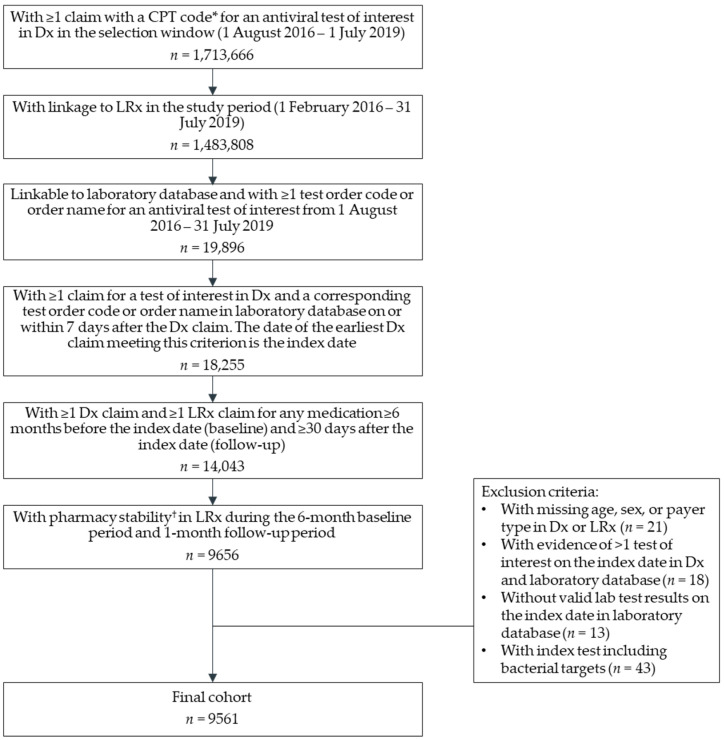
Patient selection. Abbreviations: CPT, current procedural terminology; Dx, medical claims database; LRx, prescription claims database. * CPT codes included 87,502, 87,631, 87,632, and 87,633. ^†^ A patient is considered to have pharmacy stability if ≥1 visited pharmacy consistently supplies data for the 6-month baseline and 1-month follow-up period (i.e., 7 months of stability).

**Figure 2 antibiotics-11-01058-f002:**
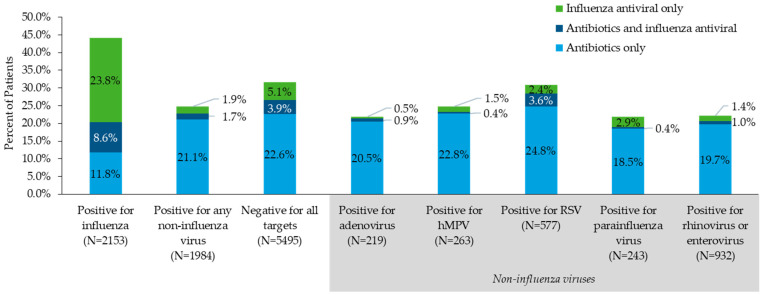
Utilization of antibiotics and influenza antivirals on or after receipt of the index test result in the study cohort stratified by test result. Abbreviations: RSV, respiratory syncytial virus; hMPV, human metapneumovirus. Test result categories are not mutually exclusive. Subgroups of any non-influenza virus are based on tests listed in Appendix A. Influenza includes “influenza A virus”, “influenza B virus”, “influenza A subtype H1”, and “influenza A subtype H3”. RSV includes “respiratory syncytial virus”, “respiratory syncytial virus A”, and “respiratory syncytial virus B”. Parainfluenza virus includes “parainfluenza virus 1”, “parainfluenza virus 2”, “parainfluenza virus 3”, “parainfluenza virus 4”, and “parainfluenza virus, specified and not specified”. Rhinovirus or enterovirus includes “rhinovirus/enterovirus” and “rhinovirus”. Treatment utilization was not reported for the 1 patient with a positive result for “coronavirus, specified and not specified”.

**Table 1 antibiotics-11-01058-t001:** Demographic and baseline clinical characteristics for all patients (*n* = 9561).

Measures	Overall Cohort *n* = 9561
**Age (years), continuous**	
Mean (SD)	36.1 (30.8)
Median (Q1, Q3)	29 (5, 68)
**Age categories, *n* (%)**	
0–4	2150 (22.5)
5–17	2182 (22.8)
18–49	1308 (13.7)
50–64	982 (10.3)
65–74	1655 (17.3)
≥75	1284 (13.4)
**Sex, *n* (%)**	
Female	5626 (58.8)
Male	3935 (41.2)
**Geographic region, *n* (%)**	
Northeast	4125 (43.1)
South	3172 (33.2)
West	1621 (17.0)
Midwest	643 (6.7)
**Payer type, *n* (%)**	
Commercial	6751 (70.6)
Medicare, including Medicare Part D	2805 (29.3)
Medicaid	5 (0.1)
**Charlson Comorbidity Index score, *n* (%)**	
0	6608 (69.1)
1	1561 (16.3)
2	750 (7.8)
3	270 (2.8)
≥4	372 (3.9)
**Specific comorbidities, *n* (%)**	
Chronic pulmonary disease *	1188 (12.4)
Asthma	854 (8.9)
Diabetes (mild to moderate)	710 (7.4)
Diabetes with chronic complications	374 (3.9)
Any malignancy	410 (4.3)
Renal disease	339 (3.5)
Congestive heart failure	281 (2.9)
History of smoking	252 (2.6)
Peripheral vascular disease	228 (2.4)
Cerebrovascular disease	188 (2.0)
Rheumatologic disease	174 (1.8)
Dementia	70 (0.7)
Myocardial infarction	49 (0.5)
HIV/AIDS	43 (0.4)
Metastatic solid tumor	25 (0.3)
Peptic ulcer disease	25 (0.3)
Hemiplegia or paraplegia	23 (0.2)
Moderate or severe liver disease	14 (0.1)
Cystic fibrosis	10 (0.1)

Abbreviations: AIDS, acquired immunodeficiency syndrome; HIV, human immunodeficiency virus; SD, standard deviation; Q1, quartile 1; Q3, quartile 3. * Chronic pulmonary disease includes asthma, bronchiectasis, chronic obstructive pulmonary disease, interstitial lung disease, and other chronic conditions.

**Table 2 antibiotics-11-01058-t002:** Characteristics of the index test.

Measures	Overall Cohort *n* = 9561
**Timing of the index date, *n* (%)**	
During an influenza season (1 October–31 May)	8787 (91.9)
Outside an influenza season (1 June–30 September)	774 (8.1)
**Specific respiratory-related diagnosis code on the index date, *n* (%) ***	
Upper respiratory tract infection, unspecified	2115 (22.1)
Influenza	1790 (18.7)
Pharyngitis	1591 (16.6)
Acute bronchitis	760 (7.9)
Sinusitis	379 (4.0)
Pneumonia	257 (2.7)
Acute nasopharyngitis (common cold)	238 (2.5)
Otitis media	137 (1.4)
Tonsillitis	127 (1.3)
Laryngitis/tracheitis	84 (0.9)
No respiratory-related diagnosis on the index date	3872 (40.5)
**Respiratory-related diagnosis code tiers on the index date, *n* (%)**	
Tier 1 (antibiotics almost always indicated)	257 (2.7)
Tier 2 (antibiotics potentially indicated)	1984 (20.8)
Tier 3 (antibiotics not indicated)	3448 (36.1)
**Ordering provider specialty (*n*, %)**	
Primary care	4415 (46.2)
Pediatrician	3312 (34.6)
Unspecified/missing	1197 (12.5)
Other specialist	423 (4.4)
Respiratory or infectious disease specialist	214 (2.2)
**Results of the index test, *n* (%) ^†^**	
Negative for all targets	5495 (57.5)
Positive for influenza	2153 (22.5)
Positive for any non-influenza virus	1984 (20.8)
Co-infected with influenza and non-influenza virus	71 (0.7)

Abbreviations: SD, standard deviation; Q1, quartile 1; Q3, quartile 3. * Categories of respiratory-related diagnosis codes are not mutually exclusive; patients may have multiple diagnosis codes on the index date. ^†^ Index test result categories are not mutually exclusive; the *n* = 71 patients co-infected with influenza and non-influenza virus are reported under “positive for influenza” and “positive for non-influenza virus”.

**Table 3 antibiotics-11-01058-t003:** Logistic regression model for predictors of ≥1 antibiotic claim on or up to 7 days after the index test result.

Variables	Overall Cohort
	*n* = 9561
	Odds Ratio (95% CI)
**Index test result (vs. Positive for influenza only)**	
Positive for non-influenza target only	1.28 (1.09, 1.51) *
Co-infected with influenza and non-influenza target	1.15 (0.63, 2.10)
Negative for all targets	1.33 (1.17, 1.50) *
**Respiratory-related diagnosis code tiers on the index date (vs. Tier 3 [antibiotics not indicated])**	
Tier 1 (antibiotics almost always indicated)	1.70 (1.30, 2.24) *
Tier 2 (antibiotics potentially indicated)	1.50 (1.32, 1.71) *
No respiratory-specific diagnosis on the index date	1.05 (0.94, 1.17)
**Age group (vs. 18–49)**	
0–4	0.90 (0.74, 1.10)
5–17	0.79 (0.65, 0.96) *
50–64	1.27 (1.05, 1.55) *
65–74	1.40 (1.14, 1.72) *
≥75	1.27 (1.01, 1.59) *
**Female (vs. Male)**	1.00 (0.91, 1.11)
**Commercial payer (vs. Medicare or Medicaid)**	0.90 (0.76, 1.06)
**Geographic region (vs. Northeast)**	
Midwest	1.63 (1.34, 1.99) *
South	1.34 (1.20, 1.50) *
West	1.53 (1.34, 1.76) *
**Prescriber specialty (vs. Primary care)**	
Pediatrician	1.05 (0.89, 1.23)
Other/missing	0.94 (0.82, 1.07)
**CCI score (vs. 0)**	
1	1.02 (0.89, 1.16)
≥2	1.08 (0.93, 1.25)

Abbreviations: CCI, Charlson Comorbidity Index; CI, confidence interval. * Indicates *p* < 0.05.

## Data Availability

Restrictions apply to the availability of the IQVIA claims data, which were used under license for the current study, and so are not publicly available. IQVIA data are available upon reasonable request and with permission of IQVIA. Laboratory data was obtained from a third party and are not publicly available. Further details can be provided by contacting A.N.

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
