# Peer review of "Outpatient Antibiotic and Antiviral Utilization Patterns in Patients Tested for Respiratory Pathogens in the United States: A Real-World Database Study"

_antibiotics, 2022, doi:10.3390/antibiotics11081058_

Round 1

Reviewer 1 Report

See attachment

Reviewer 2 Report

Manuscript antibiotics-1831372 submitted for the special issue " Use of Large Databases Related to Infectious Diseases in Primary Care" in Section “Antibiotics Use and Antimicrobial Stewardship” of Antibiotics regards a real-world database study regarding outpatient antibiotic utilization patterns in patients tested for respiratory pathogens in the United States.

The retrospective database study aims to describe outpatient antibiotic and influenza antiviral treatment patterns following molecular-based tests for acute respiratory tract infections and to explore predictors of antibiotic utilization in clinical practice. The use of a nationally representative laboratory database linked to large pharmacy and medical claims databases in the US, is very interesting. It allows for a comprehensive view of outpatient antibiotic and influenza antiviral utilization following receipt of a molecular-based viral test. It would be also interesting to enlarge this study to other States.

However, there are some points that must be considered:

The paper was not prepared according to Antibiotics guidelines. The abstract should be a single paragraph and should follow the style of structured abstracts, but without headings. Moreover, reference style was not correct, besides there are lacking parts, such as number of pages, number of Patents.

Moreover, I wonder in which period the study was carried out. The final cohort of the 9,561 patients included in the 17 study is until July 31, 2019 (as reported in Methods and Figure 1), whereas in the abstract the authors say “from August 1, 2016-July 1, 2019.”

 Minor corrections:

Ref 8: (accessed October 15). Of which year?

Ref 9: add pages and volume

Ref 11: (accessed November 1). Of which year?

Ref 25: (accessed February 15). Of which year?

Ref 36: (accessed February 15). Of which year?

General: tier or Tier. Use always the same

I suggest adding the last paragraph of Conclusion (that could be part of the actual Discussion paragraph) which summarizes the results of the study.
